# Germination of *Acacia harpophylla* (Brigalow) seeds in relation to soil water potential: implications for rehabilitation of a threatened ecosystem

Sven Arnold, Yolana Kailichova and Thomas Baumgartl

Centre for Mined Land Rehabilitation, The University of Queensland, Brisbane, Australia

## ABSTRACT

Initial soil water conditions play a critical role when seeding is the primary approach to revegetate post-mining areas. In some semi-arid climates, such as the Brigalow Belt Bioregion in eastern Australia, extensive areas are affected by open-cut mining. Together with erratic rainfall patterns and clayey soils, the Brigalow Belt denotes a unique biome which is representative of other water-limited ecosystems worldwide. Apart from other environmental stressors, germination is governed by the water potential of the surrounding soil material. While previous studies have confirmed the high tolerance of Brigalow (*Acacia harpophylla*) seeds to a broad range of temperature and salinity, the question of how soil water potential triggers seed germination remains. In this study, we used three replicates of 50 seeds of Brigalow to investigate germination in relation to water potential as an environmental stressor. Solutions of Polyethylene Glycol (PEG 6000) were applied to expose seeds to nine osmotic water potentials ranging from soil water saturation (0 MPa) and field capacity ($-.01$ to $-.03$ MPa) to the permanent wilting point ($-1.5$ MPa). We measured germinability (number of germinated seeds relative to total number of seeds per lot) and mean germination time (mean time required for maximum germination of a seed lot) to quantify germination. Based on the empirical data of the germination we estimated the parameters of the hydrotime model which simulates timing and success of seed emergence. Our findings indicate that Brigalow seeds are remarkably tolerant to water stress, with germination being observed at a water potential as low as $-1.5$ MPa. Likewise, the average base water potential of a seed population (hydrotime model) was very low and ranged between $-1.533$ and $-1.451$ MPa. In general, Brigalow seeds germinate opportunistically over a broad range of abiotic conditions related to temperature, salinity, and water availability. Direct seeding and germination of native plants on post-mining land may be an effective and economically viable solution in order to re-establish plant communities. However, due to their capacity to reproduce asexually, alternative rehabilitation approaches such as transplantation of whole soil-root compartments may become attractive for restoration ecologists to achieve safe, stable, and non-polluting ecosystems.

Corresponding author
Sven Arnold, s.arnold@uq.edu.au

## INTRODUCTION

The Brigalow Belt is an extensive bioregion located between the subtropical coastline and semi-arid interior of eastern Australia (*Arnold et al., in press*). Climatically and pedologically the bioregion is distinct from surrounding areas (*Isbell, 2002*; *Lloyd, 1984*), and as a consequence, its ecological and biodiversity attributes are unique worldwide (*Johnson, 1980*). Rainfall events occur erratically, both between years and intra-seasonally, and the clayey soils have high fertility and good water holding capacities (*Gunn, 1984*). Of the few plant species able to survive under these water-limited constraints (*Connor & Tunstall, 1968*), *A. harpophylla* (Brigalow) is the predominant one, from which the bioregion derives its name.

Since the 1950s, the delicate balance between soils and the native plant communities has been disrupted (*Eyre et al., 2009*) through clearance of Brigalow woodland for agricultural land use (cropping, grazing), which reduced the original extension of the bioregion dramatically (*Arnold et al., in press*). Consequently, *A. harpophylla* dominant ecosystems have been listed as endangered under both state (*Vegetation Management Act 1999*) and national legislation (*Environment Protection and Biodiversity Conservation Act 1999*: EPBC Act) (*Environment Australia, 2001*). More recently, areas in the Bowen Basin (largest coal reserve in Australia) have been concurrently affected by surface mine developments. The legislative requirement to reclaim post-mining land to provide safe, stable, and non-polluting environments (*Commonwealth of Australia, 2006*) provides the opportunity to re-establish native Brigalow plant communities.

Re-vegetation typically comprises either passive regeneration or proactive management such as transplanting seedlings and mature stands (*Musselman et al., 2012*), tube stocking, or direct seeding. The latter may be an effective and economically viable solution in order to re-establish Brigalow plant communities (*Engel & Parrotta, 2001*; *Lamb, Erskine & Parrotta, 2005*; *Reichman, Bellairs & Mulligan, 2006*). In this regard, seed germination is an important indicator of re-vegetation success as it represents the first stage of plant physiological response and development to environmental conditions. Seed germination can be controlled by a range of abiotic stressors of which temperature and salinity have only limited impact on *A. harpophylla* seeds within the ranges of 15–38°C and 0–20 dS/m, respectively (*Reichman, Bellairs & Mulligan, 2006*). However, given the unique pedological and climatic conditions of the Brigalow Belt, the question remains how Brigalow seeds respond to changes in water availability from soil. This quantitative information is crucial for simulating timing and success of seed emergence, for example when applying the hydrotime model (*Bradford, 1990*; *Gummerson, 1986*).

Although being the major environmental stressor in the Brigalow Belt Bioregion, less information exists on the direct effect of soil water potential on germination of *A. harpophylla* seeds. Therefore, the primary aim of this study is to determine their germination response in relation to soil water potential. In this regard, we investigate whether reduced germination at high values of soil salinity is due to toxic rather than osmotic effects (*Reichman, Bellairs & Mulligan, 2006*). As a secondary goal and based on

**Peer**J

the empirical data of the germination trials we estimate the parameters of the hydrotime model for *A. harpophylla*. Finally, we discuss our findings in the context of previous seed germination trials and alternative opportunities to re-establish plants on post-mining areas under water limited conditions.

## MATERIALS AND METHODS

### Experimental design

We used seeds[1] of *A. harpophylla* to investigate germination in relation to water potential as environmental factor. Three replicates of 50 seeds per treatment (i.e., water potential) were placed at equal distance on an absorbent substrate (Wettex®). Materials (e.g., tweezers, Wettex®, glassware) were autoclaved for 20 min at 121°C and preparation of treatments as well as any monitoring took place in a laminar flow cabinet. All treatments and replicates were then placed randomly within a germination cabinet under constant temperature (25°C) and a 12 h day and night cycle. Seeds were removed from petri dishes once a perceptible radicle emerged. The experiment ceased after five days with no further germination. No pre-treatment of seeds was required for the selected species to break dormancy (*Schmidt, 2000*; *Turnbull & Martensz, 1982*).

For controlled experiments on seed-soil relations the osmotic water potential can be used to represent soil matric potential (*Carpita et al., 1979*; *Gray, Steckel & Hands, 1990*; *McWilliam & Phillips, 1971*). We used solutions of polyethylene glycol (PEG 6000) to expose seeds to nine osmotic water potentials: 0, −.01, −.03, −.1, −.25, −.5, −.75, −1, and −1.5 MPa. These values capture soil water conditions ranging from water saturation (0 kPa) and field capacity (−.01 to −.03 MPa) to the permanent wilting point (−1.5 MPa). The empirical equation derived by *Michel & Kaufmann (1973)* and revised by *Wood, Dart & So (1993)* was used to set up the required water potential ($\psi$ in kPa):

$$\psi = (6.3 \times 10^{-5}T - 0.021 \times 96)O^{2.2357}, \tag{1}$$

where $T$ is the temperature in K (here 298.15 K), and $O$ denotes the osmolality in g 1000 g$^{-1}$ of water. Solutions measuring 15 ml of PEG 6000 were added to the seeds on the Wettex® substrate within a 90 mm petri dish and wrapped with Parafilm® to prevent evaporation. The dishes were opened each day to provide aeration and to count and remove germinated seeds.

### Germinability and mean germination time

We used germinability $G$ (%) and mean germination time $\bar{t}$ (days) as measurements to quantify germination (*Ranal & Santana, 2006*). While $G$ simply represents the number of germinated seeds ($g$) relative to the total number of seeds per replicate ($n$):

$$G = (g/n)\,100, \tag{2}$$

$\bar{t}$ denotes the mean length of time required for maximum germination of a seed lot (*Czabator, 1962*):

$$\bar{t} = \sum_{i=1}^{k} g_i t_i \bigg/ \sum_{i=1}^{k} g_i, \tag{3}$$

[1] Seeds were collected in November 2012 from the native Brigalow catchment at the Brigalow Research Station (*Cowie, Thornton & Radford, 2007*).

**Table 1 Osmotic pressure $\psi_o$ and NaCl concentration based on the electrical conductivity used in *Reichman, Bellairs & Mulligan (2006)*.**

| Electrical conductivity (dS m$^{-1}$)[a] | NaCl concentration (g kg$^{-1}$ H$_2$O)[b] | Osmotic pressure $\psi_o$ (MPa) |
|---|---|---|
| 0 | 0 | 0 |
| 5 | 1.17 | $-.05$ |
| 10 | 4.57 | $-.195$ |
| 15 | 6.86 | $-.293$ |
| 20 | 9.14 | $-.391$ |
| 25 | 11.43 | $-.489$ |
| 30 | 13.7 | $-.586$ |

[a] *Reichman, Bellairs & Mulligan (2006)*.
[b] *United States Salinity Laboratory Staff (1954)*.

where $t_i$ is the time elapsed from initiation of the experiment to the $i$th observation day until germination ceases on the $k$th day.

We applied a generalised logistic regression model (GLM) on a logit scale and simple linear regression for $G$ and $\bar{t}$, respectively, to test the significance of the relationship between germination percentage or time required for maximum germination and decreasing water potential. Assumptions were tested via diagnostic plots, which showed no violation of homogeneity of variance and normal distribution for the residuals. Separate GLM models were adopted for each species as this provided the best fit for the data. We also employed other models (linear regression and logistic regression with a probit scale) and various transformations (arcsine, square root transformations) but they either did not fit the data, or when they did, failed homogeneity of variance and normal distribution of error ($p > 0.05$).

We compared the results for $G$ with those published by *Reichman, Bellairs & Mulligan (2006)* for salinity. Therefore, we estimated the osmotic pressure $\psi_o$ (MPa) based on the value range of electrical conductivity used in *Reichman, Bellairs & Mulligan (2006)*:

$$\psi_o = MRT, \qquad (4)$$

where $M$ is the molarity (mol L$^{-1}$) of the sodium chloride solution, $T$ is the temperature in K (here 298.15 K), and $R$ is the gas constant (J mol$^{-1}$ K$^{-1}$). The corresponding values of $\psi_o$ are presented in Table 1.

## Estimating parameters of the hydrotime model

The hydrotime model was proposed by *Bradford (1990)*, *Gummerson (1986)* to "*simultaneously account for both the timing and the extent of germination of a given seed population in relation to its $\psi$ environment*" (*Bradford, 2002*). It is defined as:

$$\theta_H = (\psi - \psi_b(P))t_P, \qquad (5)$$

where $\theta_H$ is the hydrotime constant (MPa h), $t_P$ (h) is the time to germination of percentage $P$ of the seed population, and $\psi_b(P)$ (MPa) is the base or threshold water

potential above which $P$ can still complete germination. Among the seed population, $\psi_b$ is variable and can be described by a range of frequency distributions (*Mesgaran et al., 2013*) of which the normal frequency distribution is the most commonly used one defined by its mean $\psi_b(P_{50})$ and standard deviation $\sigma_{\psi b}$ (*Gummerson, 1986*). Together, these parameters enable prediction of the germination time courses, i.e., both rate and extent of germination of the seed population, at any $\psi$ at a constant temperature (*Bradford, 2002*):

$$t_P = \theta_H / (\psi - \psi_b(P_{50})). \tag{6}$$

In this regard, $\theta_H$ "*quantifies the inherent speed of germination, which can vary among species and physiological states*" (*Bradford, 2002*), whereas $\psi_b(P_{50})$ indicates the average stress tolerance of a seed population, and $\sigma_{\psi b}$ describes the synchrony in germination timing among seeds in a population. We estimated these parameters based on the observed values of $t_P$ for each $\psi$, i.e., we plotted the germination rates ($1/t_P$ in $h^{-1}$) as a function of $\psi$ and fitted regressions to the observed data (Fig. 4). This resulted in straight lines with slopes of $\theta_H^{-1}$ and intercepts on the $\psi$-axis corresponding with $\psi_b(P)$. The values of $\psi_b(P)$ were then fitted to a normal and log-logistic frequency distribution, respectively, and their distribution parameters were estimated accordingly.

## RESULTS

Spearman's correlation coefficient of $-0.96$ indicated a significant ($p = 10^{-4}$) negative correlation between mean time required for maximum germination and germinability (Fig. 1). Above $-.75$ MPa germinability was remarkably large and mean germination time small. The mean time required for maximum germination ranged between three and five days at water potentials higher than $-.75$ MPa and increased to over seven days at water potentials lower than $-1$ MPa (Fig. 2B). Likewise, germinability was highest ($90\% \pm 9\%$) at the water potential corresponding to saturated soil water conditions (0 MPa) and significantly decreased with decreasing water potential ($p = 0.003$) to $65\% \pm 11\%$ and $20\% \pm 9\%$ at water potentials of $-.75$ MPa and $-1$ MPa, respectively (Fig. 2A). Remarkably, at a water potential as low as $-1.5$ MPa (permanent wilting point) still $4\% \pm 1.8\%$ of the seeds germinated. Accordingly, the estimated mean base water potentials were $-1533$ and $-1451$ MPa for the normal and log-logistic distribution, respectively (Fig. 3), and the estimated hydrotime constant was $.0607$ MPa h (Table 2).

In Figure 4 we plotted the germinability of Brigalow seeds in relation to the osmotic pressure based on solutions of sodium chloride as applied by *Reichman, Bellairs & Mulligan (2006)*, and solutions of Polyethylene Glycol as applied in this study. While no significant differences ($p < 0.05$) were observed between the two treatments for osmotic pressures above $-.4$ MPa, germinability was smaller under the sodium chloride treatment for osmotic pressures below $-.4$ MPa.

## DISCUSSION

The findings of this study allude to the seed water condition of *A. harpophylla*. Together with the initial soil water conditions, these state variables have crucial implications for the rehabilitation of post-mining areas under water-limited conditions.

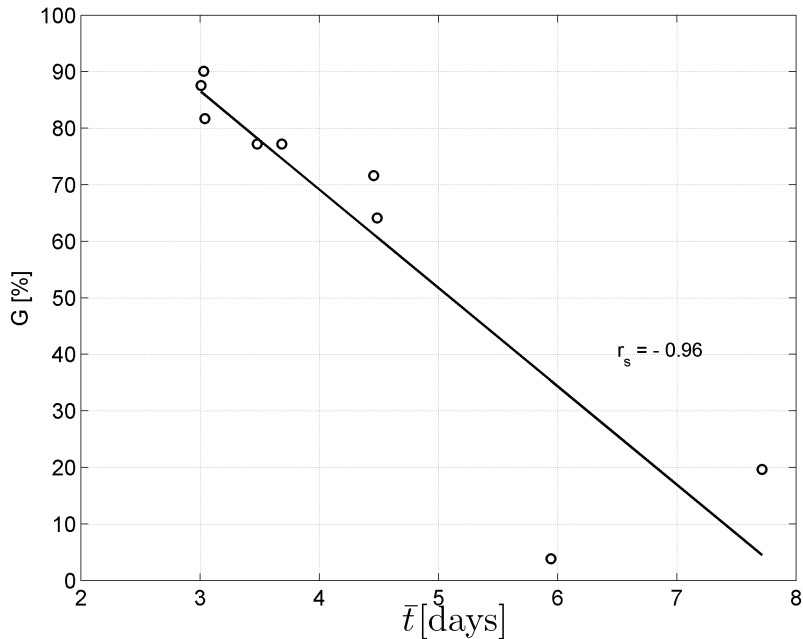

**Figure 1  Negative correlation between mean germination time and germinability.** Spearman's correlation coefficient $r_s$ is significantly different from zero ($p = 10^{-4}$).

**Table 2  Osmotic pressure and NaCl concentration based on the electrical conductivity used in *Reichman, Bellairs & Mulligan (2006)*.** Parameters estimates of the the hydrotime constant (*Bradford, 2002*), and the log-logistic distribution (scale $\alpha$, shape $\beta$) and normal distribution (mean $\mu$, standard deviation $\sigma$) based on the germination trials of *A. harpophylla*.

| Parameter | Estimated value | Standard error |
|---|---|---|
| $\theta_H$ (MPa h) | .0607 | .0117 |
| | *Normal distribution* | |
| $\mu$ (MPa) | $-1.533$ | .798 |
| $\sigma$ (MPa) | .642 | .257 |
| | *Log-logistic distribution* | |
| $\alpha$ (MPa) | $-1.451$ | .0012 |
| $\beta$ ($-$) | 1.28 | 1.09 |

## Soil and seed water conditions

Initial soil water conditions play a critical role when direct seeding is the primary approach to revegetating post-mining areas. This is even more important in a semi-arid climate, where water is limited due its scarcity or erratic occurrence (*Rodriguez-Iturbe & Proporato, 2004*). Seed germination is triggered by the amount of water the seed can imbibe, which is related to the water potentials of both the soil and the seed (*Bewley et al., 2013*; *Bradford, 2002*; *Evans & Etherington, 1990*; *Williams & Shaykewich, 1971*). In this regard, water can only enter the seed if the seed water potential is below the water potential of the surrounding soil material. In general, air-dry seeds have water potential

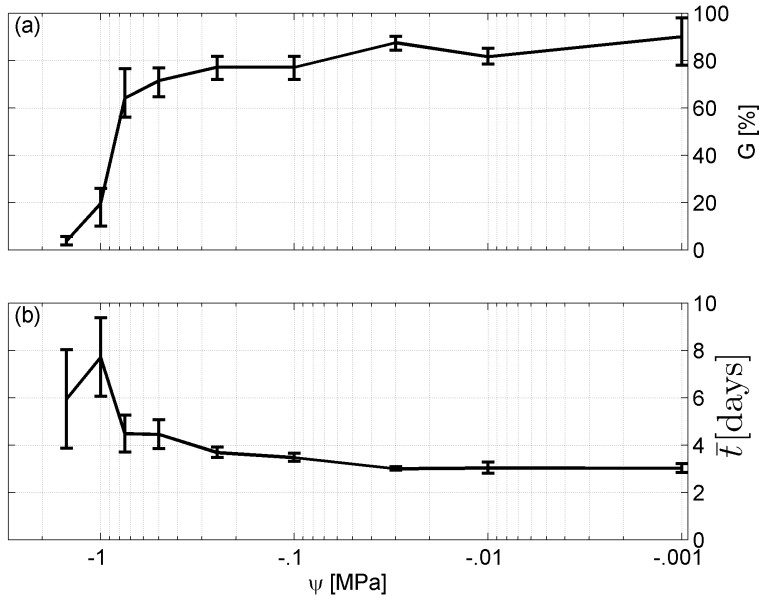

**Figure 2  Germination results.** (A) Germinability G and (B) time required for maximum germination $\bar{t}$ of *A. harpophylla* in relation to the water potential $\psi$. Error bars indicate the standard deviation across 3 replicates of 50 seeds.

[2] $\psi_b(P_{50}) = -1.39$ to $-.59$ MPa (based on 36 species)

[3] $\psi_b(P_{50}) = -1.42$ to $-1.13$ MPa (based on 4 species)

values of $-50$ to $-350$ MPa (*Bewley et al., 2013*). The empirical results of this study indicate that seeds of *A. harpophylla* are very water stress tolerant, that is, their seed water potential is extraordinarily low. For example, radicles still emerged at a water potential as low as $-1.5$ MPa (Fig. 2A). Likewise, the mean base water potential of the seed population ranges between $-1.533$ and $-1.451$ MPa (Table 2 and Fig. 3). These are remarkable values compared with selected Mediterranean (*Köchy & Tielbörger, 2007*)[2] and agricultural plant species (*Watt, Bloomberg & Finch-Savage, 2011*)[3] . Qualitatively, the low seed water potential corresponds well with other investigative studies on physiological water relations of *A. harpophylla* (*Doley, 2004*), which measured foliage water potentials as low as $-15$ MPa (*Connor & Tunstall, 1968*) and shoot water potentials of $-7.2$ to $-6.8$ MPa (*Connor, Tunstall & Van den Driessche, 1971*; *Tunstall & Connor, 1981*; *Van den Driessche, Connor & Tunstall, 1971*). Also the seed coat, which is atypically soft compared with other Acacia species, denotes an adaptive mechanism to erratic rainfall patterns of the Brigalow Belt Bioregion to rapidly overcome dormancy if soil water conditions are elevated (*Johnson, 1964*; *Reichman, Bellairs & Mulligan, 2006*; *Scott, Jones & Williams, 1984*).

The parameter estimates of the hydrotime model (Table 2) play a critical role for predicting the time required to germinate a fraction of the seed population under given soil water conditions (Eq. (5)). In this regard and together with the physiological parameters of the hydrotime model (i.e., hydrotime constant and distribution of the base water potential) the soil water potential governs germination and, consequently, the

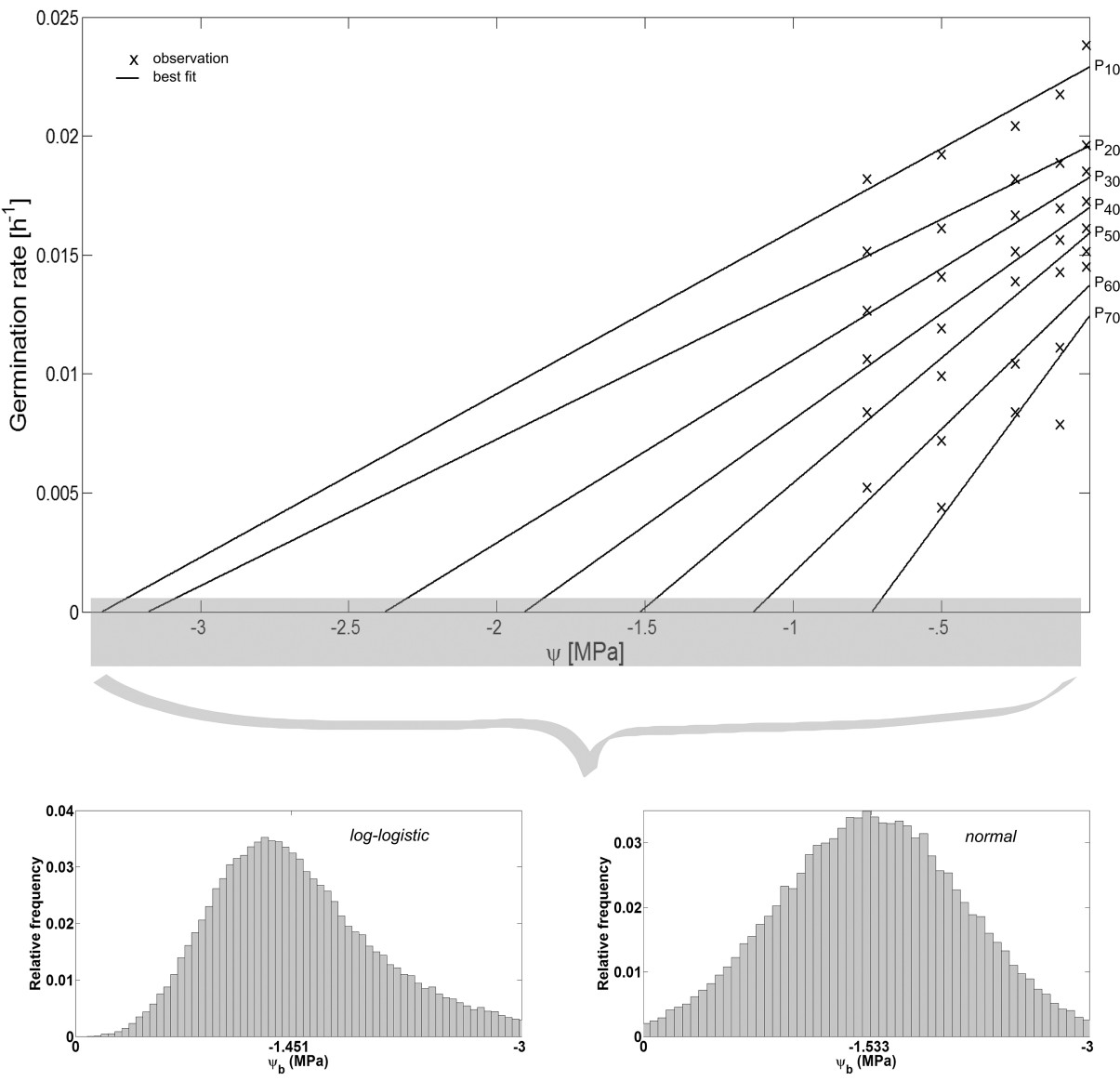

**Figure 3 Hydrotime model-parameters.** Germination rate in relation to water potential $\psi$ (adapted according to *Bradford (2002)*). While the slopes of the curves correspond to the reciprocal of the hydrotime constant $\theta_H^{-1}$, the interceptions with the $\psi$-axis correspond to the base water potential $\psi_b$ for a given percentage $P$ of the seed population. The values of $\psi_b$ can be described by a range of frequency distributions (*Mesgaran et al., 2013*) among the seeds of a lot (inset). Parameter estimates of the log-logistic ($\alpha, \beta$) and normal distribution ($\mu, \sigma$), and $\theta H$ are presented in Table 2.

success of initial vegetation recruitment and early ecosystem establishment. Moreover, together with empirical data on the effect of temperature on the germination of Brigalow seeds (*Reichman, Bellairs & Mulligan, 2006*), the findings of this study can be utilised to parameterise hydrothermal models (*Bradford, 2002*; *Bullied, Van Acker & Bullock, 2012*; *Gummerson, 1986*; *Köchy & Tielbörger, 2007*; *Watt, Bloomberg & Finch-Savage, 2011*) for

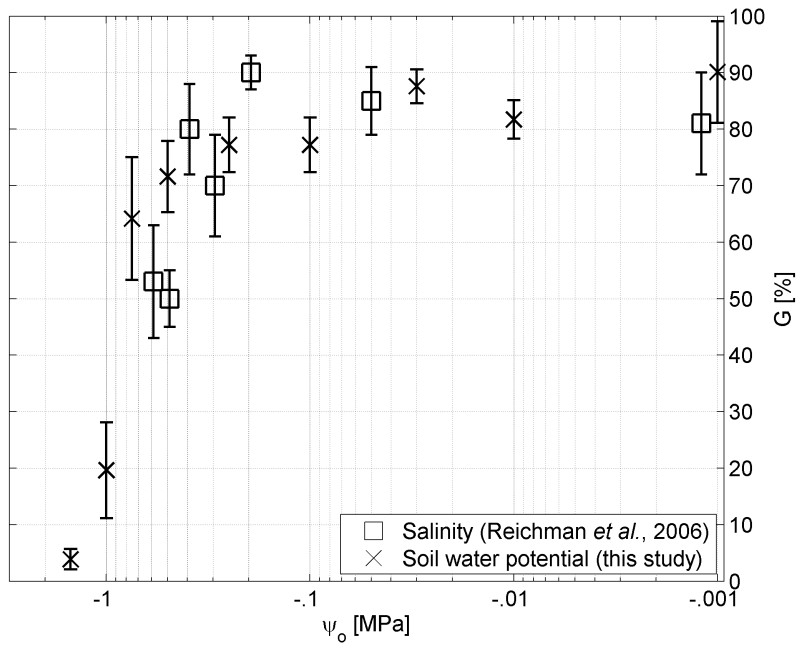

**Figure 4  Water potential vs salinity.** Germinability *G* of *A. harpophylla* in relation to the osmotic pressure $\psi_o$ based on solutions of sodium chloride (squares (*Reichman, Bellairs & Mulligan, 2006*)) and PEG (crosses). Error bars indicate the standard error across 4 replicates of 20 seeds (*Reichman, Bellairs & Mulligan, 2006*) and 3 replicates of 50 seeds for salinity and soil water potential, respectively.

predictive modelling of germination in relation to the two environmental factors of temperature and water availability.

The results of this study show that germinability decreases and the time required for maximum germination increases with decreasing soil water potential (Fig. 2). Moreover, the strong negative correlation between germinability and mean germination time (Fig. 1) underpins how important a rather short germination time is to maximise germination success. Thus, topsoil restoration at post-mining land in the Brigalow Belt Bioregion should target to maximise the initial soil water potential while explicitly considering the erratic character of rainfall patterns in Central Queensland (*Audet et al., 2013*; *Audet et al., 2012*). This can be accomplished by optimising soil attributes such as depth, texture, compaction, organic amendments, etc. (*Arnold, 2012*; *Arnold et al., in press*; *Zipper et al., 2013*). Apart from physical soil restoration, pre-treatment of seeds, i.e., seed priming, also plays a critical role to enhance germination (*Jisha, Vijayakumari & Puthur, 2013*). Amongst the broad range of seed priming techniques, hydropriming and osmopriming are the most promising approaches for plant establishment in semi-arid climate to increase seedling growth (*Yagmur & Kaydan, 2008*), and root and shoot length (*Kaur, Gupta & Kaur, 2002*). Although the exact mechanisms behind pre-treatments of seeds are not fully understood, seed priming seems to activate cell signalling pathways and cellular responses to environmental stressors resulting in faster plant defence responses (*Jisha, Vijayakumari & Puthur, 2013*).

**Peer**J

[4] $G = 90\%$ after 14 days.

Despite plant available water being the primary environmental factor in the Brigalow Belt Bioregion (*Arnold et al., in press*), due to hot summers and highly saline mine spoils secondary stressors such as temperature and salinity may also play a critical role for the germination success of *A. harpophylla*, which was tested by *Reichman, Bellairs & Mulligan (2006)*. Their findings indicate no significant trend in germination among the temperature range of 15–38°C.[4] Likewise, seeds of *A. harpophylla* showed remarkable tolerance to salinity up to an electrical conductivity of 30 dS m$^{-1}$, however, with significant reduction in germination at salinity greater than 20 dS m$^{-1}$. The authors speculated "*it seems unlikely that the reduced germination at 25 and 30 dS/m were due to osmotic effects*" (*Reichman, Bellairs & Mulligan, 2006*). While the present study confirms this conclusion for high values of salinity (Table 1) corresponding to osmotic pressure values greater than $-.4$ MPa (Fig. 4), no significant differences could be found between treatments of sodium chloride and PEG (Materials and Methods) for small values of osmotic pressure corresponding to high salinity (Fig. 4). That said, we conclude that under low to moderate levels of salinity the osmotic pressure plays the primary role for germination of *A. harpophylla* seeds rather than the toxic nature of the predominant salt (here sodium chloride), whereas salinity becomes the primary environmental stressor under high salt concentrations. More generally, seeds of *A. harpophylla* seem to germinate quite uniformly over a broad range of environmental conditions related to temperature, salinity, and water availability.[5]

[5] Note that soil hydraulic conductivity can become a limiting factor in dry soils (*Bewley et al., 2013*).

## Alternative rehabilitation approaches

Despite the opportunistic germination capability of *A. harpophylla* seeds, the question remains whether direct seeding denotes the optimal approach to rehabilitate native Brigalow ecosystems given the hydropedological,[6] climatic, and plant physiological attributes and conditions of the Brigalow Belt Bioregion.

[6] Interface between the pedosphere and the hydrosphere (*Li, Lin & Levia, 2012*)

Hydropedological processes are fundamental for the proliferation of Brigalow plant communities (*Arnold et al., in press*). The predominant clay soils (*Isbell, 2002*) comprise fine-textured non-cracking Grey and Black Dermosols (Lixisols (*Rees et al., 2010*; *IUSS Working Group WRB, 2006*)), and uniform dark cracking Grey and Black Vertosols (Vertisols (*Rees et al., 2010*)). The latter form "gilgais" (*Cowie, Thornton & Radford, 2007*; *Radford et al., 2007*; *Thornton et al., 2007*), which denote ephemeral water storages if filled during intensive storm events. The climate is characterised by erratically distributed rainfall patterns with short and intensive storm events occurring during summer (*Cowie, Thornton & Radford, 2007*), associated with the risk of water logging or soil erosion; whereas the very arid conditions during the winter season generally involve periods of water deficit (*Audet et al., 2013*; *Audet et al., 2012*; *Dalton, 1993*). In combination with these erratic climatic conditions the soil properties facilitate rather low soil water potentials on the long-term (*Tunstall & Connor, 1981*), and as a consequence, these landscapes can be colonised by a few plant species only, which is reflected in relatively low species richness and total vegetative biomass (*Isbell, 2002*; *Johnson, 1980*). As elaborated in the previous section, among the species in the Brigalow Belt Bioregion, *A.*

*harpophylla* developed very well adapted mechanisms to proliferate under these harsh conditions. The plants flower only sporadically (*Benson et al., 2006*) and thus seeds and seedlings are only produced in large numbers during years of extraordinary rainfall (*Butler, 2007*; *Johnson, 1997*). However, *A. harpophylla* is able to reproduce asexually, i.e., through root suckering or sprouting even if aboveground parts of the plant are damaged dramatically, as long as belowground biomass and hydropedology stay intact (*Arnold et al., in press* and references therein). The extent of this vegetative reproduction is most pronounced if trees are young and severely damaged, and under dry conditions (*Johnson, 1964*). In Colorado, USA circumstances are similar with regard to the re-establishment of vegetatively regenerating Aspen (*Populus tremuloides*) on surface-mined land (*Musselman et al., 2012*), which initially failed due to severe damage to the root system and thus the limited access to water and nutrients (*Shepperd & Mata, 2005*). However, rehabilitation was more successful when more comprehensive soil-root compartments were transplanted from local sources in combination with weed control and light irrigation with non-saline water (*Musselman et al., 2012*). In this regard, it seems to be crucial to keep the delicate balance between soil attributes (depth, compaction, texture) and root extension in balance. This innovative rehabilitation approach of transplanting the whole soil-root compartment may also be of interest for restoration ecologists engaged with re-establishment of Brigalow ecosystems in Central Queensland to achieve safe, stable, and non-polluting ecosystems (*Commonwealth of Australia, 2006*). However, the advantage of direct seeding over asexual propagation such as transplantation is the gain of genetic diversity – critical if ecosystems are forced to adapt to the projected changing climate of Central Queensland (*Low, 2011*).

## ACKNOWLEDGEMENTS

The authors would like to thank Steve Adkins and David Doley for critical discussions, as well as the two reviewers Ana Vasques and Teresa Eyre for constructive comments. Further, we thank Stuart Irvine-Brown for collection of *A. harpophylla* seeds – a pivotal contribution to this study.

### Funding

Sven Arnold was kindly supported by the Postdoctoral Research Fellowship Scheme and the Early Career Research Grant of The University of Queensland. The funders had no role in study design, data collection and analysis, decision to publish, or preparation of the manuscript.

### Grant Disclosures

The following grant information was disclosed by the authors:
The Postdoctoral Research Fellowship Scheme
Early Career Research Grant of The University of Queensland.
## Competing Interests

The authors declare that they have no competing interests.

## Author Contributions

- Sven Arnold conceived and designed the experiments, analyzed the data, contributed reagents/materials/analysis tools, wrote the paper.
- Yolana Kailichova performed the experiments, analyzed the data, contributed reagents/materials/analysis tools.
- Thomas Baumgartl conceived and designed the experiments, providing leadership.

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
