# Peer review of "Germination of Acacia harpophylla (Brigalow) seeds in relation to soil water potential: implications for rehabilitation of a threatened ecosystem"

_PeerJ, doi:10.7717/peerj.268_

## Round 0.1 · original submission · Major Revisions

I am pleased to inform you that your paper has been reviewed favourably, indicating that some significant improvements are needed before the paper can be accepted for publication.

·

Basic reporting

The article is written in correct and clear English, having an adequate structure and the background is well explored. The research is relevant and meaningful and has an important advantage of having an experimental setup that allows the estimation of parameters for modelling.
Further improvements include the following:
The units of water potential used could be changed to MPa (instead of KPa) to improve the potential comparison of the results attained with other current germination studies.
The figure indexing should follow the order in which they are mentioned in the text (Results section).
The interpretation of the germination percentages attained under the lower water potential tested (-1.5 MPa) in comparison to water potentials present in other parts of the plant should be adjusted (lines 140-143) because seeds can have very low internal water potentials (please see Bewley et al. 2013). Furthermore, these interpretations should be left for the discussion section, being excluded from the result section (lines 115 and 116).

Experimental design

The objectives should be clarified through a more specific research question/hypothesis. To further clarify the hypothesis underlying this study I suggest that the expected germination response under polyethylene and saline solutions (addressed in the discussion section) is also addressed in the objectives of the study.

Validity of the findings

The implications of the present findings for the practice of rehabilitation of post-mining areas should be further explored including the discussion of the use of seed priming in germination enhancement. Other aspects for the improvement of germination predictions, such as the use of combined models of temperature and water potential (hydrothermal models: lines 163 and 164) should also be mentioned.
Lines 171-175: the importance of hydraulic conductivity could also be addressed (Bewley et al. 2013).
Section 4.2.: The discussion of the adequacy of seeding for the rehabilitation of native Brigalow ecosystems should be further clarified giving the pros and contras of this approach in contrast with the use of asexually propagation methods. Other aspects of both approaches, such as genetic diversity could also be discussed.

Additional comments

Bewley, J. D., Bradford, K. J., Hilhorst, H. W., & Nonogaki, H. (2013). Seeds, Physiology of Development, Germination and Dormancy, pp 140-147

·

Basic reporting

The submission appears to adhere to PeerJ templates and policies.
A few editing suggestions...

Abstract, 1st line, change 'of' to 'to'; line 4, insert comma after 'soils'; line 5, change 'for' to 'of'; line 6, insert comma after 'stressors'; line 19, delete 'predominant native'; line 11 insert 'an' between 'as' and 'environmental'; line 17, delete 'facilitates'; line 18, insert 'tolerant' between 'remarkably' and 'water'; line 19, insert comma after 'stress' and delete 'tolerant'; line 25, delete 'ability of asexual reproduction' and insert 'capacity to reproduce asexually' instead.

Introduction, line 16: delete 'bioregion' and replace with 'A. harpophylla dominant ecosystem'. The bioregiob us bit kusted as endangered, the ecosystem is.

Discussion, line 130; replace 'under' with 'in a'.

Experimental design

The experimental design appears sound. My only query - relevant to section 2.1, line 43 of the MS, regards where the Acacia harpophylla seeds were sourced from. Where they sourced in the field? or elsewhere.

Validity of the findings

Findings are highly valid, and data robust and analyses statistically sound. As mentioned earlier, I would like to see stated in the MS where the seeds were sourced from.

Additional comments

This is a very interesting article on a little known topic - the germination of Acacia harophylla seeds. Acacia harpophylla communities are of conservation concern both at Australian state and federal levels, so the restoration of these communities is of particular interest. The authors have put together a succinct and well written and scientifically sound paper, and should be congratulated.

---

## Round 0.2 · accepted · Accept

The reviewers comments and remarks have been addessed carefully by the authors, and the current version of the manuscript is now acceptable for publication in PeerJ.